# The Functional Power of the Human Milk Proteome

**DOI:** 10.3390/nu11081834

**Published:** 2019-08-08

**Authors:** Jing Zhu, Kelly A. Dingess

**Affiliations:** 1Biomolecular Mass Spectrometry and Proteomics, Bijvoet Center for Biomolecular Research and Utrecht Institute for Pharmaceutical Sciences, University of Utrecht, Padualaan 8, 3584 CH Utrecht, The Netherlands; 2Netherlands Proteomics Center, Padualaan 8, 3584 CH Utrecht, The Netherlands

**Keywords:** human milk, protein, glycoprotein, endogenous peptide, breastfeeding

## Abstract

Human milk is the most complete and ideal form of nutrition for the developing infant. The composition of human milk consistently changes throughout lactation to meet the changing functional needs of the infant. The human milk proteome is an essential milk component consisting of proteins, including enzymes/proteases, glycoproteins, and endogenous peptides. These compounds may contribute to the healthy development in a synergistic way by affecting growth, maturation of the immune system, from innate to adaptive immunity, and the gut. A comprehensive overview of the human milk proteome, covering all of its components, is lacking, even though numerous analyses of human milk proteins have been reported. Such data could substantially aid in our understanding of the functionality of each constituent of the proteome. This review will highlight each of the aforementioned components of human milk and emphasize the functionality of the proteome throughout lactation, including nutrient delivery and enhanced bioavailability of nutrients for growth, cognitive development, immune defense, and gut maturation.

## 1. Introduction

Human milk via breastfeeding is the gold standard for infant feeding, as it provides not only nutritional excellence, but also protective effects during a time of unmatched antigenic and pathogenic challenges. Both short and long term health benefits have been attributed to breastfeeding by clinical and epidemiological studies [1]. Some short and long term benefits include, but are not limited to, lower incidences of diarrhea, respiratory and urinary tract infections, and otitis media; reduction of disease risk, such as, asthma, allergy, and type I and II *diabetes mellitus* [2,3,4,5]. Additionally, it has been shown to have protective effects against the development of noncommunicable diseases commonly associated with inflammatory pathogenesis, such as obesity and cardiovascular disease [2,3,4,5]. The act of breastfeeding has also shown to be beneficial for maternal health, including reduced risk for development of rheumatoid arthritis, cardiovascular disease, diabetes, and breast and ovarian cancer [1]. For all these reasons and more, exclusive breastfeeding for six months, and in conjunction with complementary food feeding through one year of life or longer, as desired by the mother and infant, are recommended by the American Academy of Pediatrics and the World Health Organization [1]. 

The human milk proteome is comprised of not only proteins, which are highly glycosylated, but also endogenous peptides that are derived from proteins within the mammary gland maintaining their own distinct functions. One must consider the overall composition of milk, as it is one of the most complex and dynamic biofluids of the human body, to understand the function of the human milk proteome. Human milk is comprised of three major macronutrients, carbohydrates, fats, and proteins, listed by order of their relative abundance [6]. These macronutrients are continuously changing in their composition and concentration throughout lactation to meet the demanding needs of the infant during growth and development. 

In this review, we aimed to provide a comprehensive overview of the human milk proteome, which is obtained by applying multiple omics approaches, including proteomics, peptidomics, and glycoproteomics. We believe that combined analysis of multiple omics approaches will help in providing an increased understanding of the functionality of each constituent of the proteome. This review will highlight each component of the human milk proteome, proteins, including enzymes/proteases, glycoproteins, and endogenous peptides. Additionally, we will emphasize the functionality of the proteome throughout lactation, including nutrient delivery and enhanced bioavailability of nutrients for growth, cognitive development, immune defense, and gut maturation.

## 2. Factors that Affect Milk Composition

The lactational stage is one of the key designators for determining the composition of human milk, which is traditionally based on day postpartum and further categorized as colostrum (< 72 h postpartum), transitional (> 3–15 days), and mature (> 16 days) milk [1,6,7]. These stages of lactation are more accurately categorized by the maturity of the mammary gland and the functions of the proteome it is producing. Colostrum is produced in the lactational stage when the protein concentration is highest, especially regarding immune modulatory proteins, as the mammary gland is still maturing. Typically, the maturity of the mammary gland involves tight junction closures of the mammary epithelium and regulation of the Na^+^:K^+^ efflux [7]. Colostrum is then followed by transitional milk, in which there is an upregulation of protein synthesis and an inhibition of protein degradation. Last is mature milk in which there is a switch from protein production to fatty acid synthesis. 

Reports indicate that breast milk composition becomes relatively stable between 2–12 weeks of lactations and that as the mammary gland matures from a transitional to a mature stage there is less protein variance [8]. Overall, while considering the entire lactational period, the protein concentration in human milk is typically in the range of 10–20 g/L [9], starting high in colostrum and then steadily declining until it reaches a plateau and then becomes designated as mature milk. Table 1 illustrates detailed protein concentrations and functionality. For generating Table 1, individual proteins were searched, articles were considered first for term milk and mass spectrometry (MS) derived data. If this was not available, then literature values based on preterm milk and or immune assays were used. Overall, we aimed to use the most current literature possible. The overall shift in milk composition is important, because it shows the adaptability of milk to meet the changing functional needs of the infant. Shifting from innate immune initiation to adaptive immune learning and becoming more energy dense to better meet the caloric needs of the growing infant [10] and their diversifying gut microbiome.

Recent research has focused on exploring the human milk proteome longitudinally and determined that the protein content of human milk changes throughout the course of lactation based on functionality, rather than by differing mothers or differing populations [11,12,13]. This change in functionality meets the changing needs of the developing infant where an early lactational stage was characterized by greater concentrations of the immunoglobulins IgA and IgM and a switch at later lactational stage was characterized by the enhanced abundance of IgG [11,12]. This data suggests that human milk transitions from a defense mechanism of the newborn and direct pathogen-killing, to a more mature milk supporting an independent immune system [11,12]. Overall, there is ample evidence that human milk helps to establish both innate and adaptive immune responses of the infant as the infant matures.

There are many factors that affect the composition of human milk, independent of the lactational stage. These factors can be maternal, such as diet, smoking status, exercise, BMI, and infant factors, such as gestational age and sex of the infant [9,10]. Additionally, the breast milk composition can fluctuate within feeds and diurnally, and it is influenced by maternal diet [10]. Recent evidence has shown that foreign proteins, from bovine milk, can be detected in human milk, and that the most likely source for these foreign proteins is via the maternal diet [14]. Diurnal changes have been identified in the fat composition of milk, but such changes are not well investigated or reported for proteins, protein post-translational modifications (PTMs), and/or endogenous peptides. Additionally, the overall proteome can be impacted by inhibitors and activators, present within the mammary gland, and throughout the body allowing for constituents of the proteome to be translocated to the mammary gland via circulation, as described in more detail in Section 2.1. Future research should aim to explore the diurnal changes of the human milk proteome and all of its components. 

Human milk cannot be artificially mass-produced in all details of its natural complexity, instead it is tailor made to meet the precise and unique needs of each mother-infant pair. Moreover, it has undergone more than 300 million years [15] of evolutionary selective pressure to become this intricate, dynamic, and essential element of human nutrition. The proteome is an exemplary model of this, as it can reveal the complex differences between individuals while meeting the functional needs of the infant. Wherein the biological functionality of human milk is determined by a series of interacting and synergistic factors that are conserved, regardless of the considerable qualitative and quantitative differences in the human milk proteome between individuals.

### 2.1. Mammary Gland Physiology

The lactating mammary gland, which is comprised of branching networks of ducts formed from epithelial cells making up extensive lobulo-alveolar clusters, is the site of milk production and secretion. These lactocytes are responsible for 80–90% of the protein content of human milk, with the remaining percentage coming from maternal circulation [10]. Proteins and endogenous peptides enter the inner luminal space of the mammary gland through four general pathways (Figure 1), belonging to either transcellular or paracellular routes that were extensively summarized by other reviews [26,34,35,36,37]. In brief, the exocytotic pathway (pathway 1) is the dominant way for the secretion of endogenously generated proteins, including the major milk proteins, e.g., caseins, α-lactalbumin, and lactoferrin. This secretion mechanism is similar to exocytotic pathways that were found in other cell types [37]. Lipid-associated proteins existing in milk fat globule membrane (MFGM) are secreted by a process that is unique to mammary epithelial cells (pathway 2) [38], including mucins and enzymes [39]. Another transcellular pathway, the transcytosis pathway (pathway 3), is responsible for the transporting of proteins from serum or stromal cells, e.g., secretory immunoglobulin A (sIgA) [40], albumin [41] and transferrin [42]. A benefit of this pathway is that proteins can be released into the lumen in their intact and active forms [26]. Finally, one paracellular pathway (pathway 4), directly transports serum substances and cells [43]. This paracellular pathway is only available on special occasions, such as inflammation or preterm birth [44], and it is normally closed by the tight-junctions between epithelial cells [44,45]. 

Human milk glycoproteins follow these same four general pathways, but glycosylation can be locally impacted by the glycan biosynthesis in the mammary gland or globally throughout the body. Maternal genetic factors may also contribute to the protein glycosylation profile that is produced. Two fucosyltransferases, including fucosyltransferases 2 (FUT2) and fucosyltransferases 3 (FUT3), which are encoded by the secretor and Lewis genes, respectively, are responsible for adding the terminal fucose in an α1,2 linkage and subterminal fucose in α1,3 and α1,4 linkage [46]. Some glycans that are bound to proteins then carry distinct elongated glycan epitopes that are determined by polymorphisms [47].

The secretion and transport of endogenous peptides is impacted by protein associations and conformational availability to be digested, or not, to shorter peptides by proteases and protease inhibitors. The transport of proteases from circulation to MECs is not fully understood and much remains unanswered, such as, whether proteases pass MECs by transcellular or paracellular pathways or whether specific receptors and transports exist for proteases [26]. Literature referring to bovine milk indicates that proteases in milk are not the source of somatic cells and that these proteases are therefore derived from blood [26,48]. Maternal health also plays a role in protein degradation to peptides as infections, especially those within the mammary gland, can result in an increase in proteolysis [26]. The health of the infant can influence this same process, as there is feedback communication between the infant and the mother during breastfeeding. When the infant is sick salivary proteins can make their way up the mammary ducts via the nipple during suckeling, which then signals the transcription of immune modulating proteins within the mammary gland that are needed by the infant [49,50]. This opens up an exciting and unexplored avenue of research to better elucidate the maternal infant communication mechanism that is taking place at the mammary-oral interface. Furthermore, this could then lead to potentially novel human milk protein and or endogenous peptide biomarkers in relation to infant health. 

## 3. Proteins

Human milk, as the preferred source of infant nutrition, provides essential and non-essential amino acids via its protein and free peptide fraction, which in turn are used by the infant for protein synthesis required for growth. Therefore, the analysis of human milk proteins is important for the determination of protein requirements to meet the infant’s needs. Moreover, these needs are individual specific and they are based on the infant’s ability to utilize and break down dietary proteins.

Historically milk proteins have been classified into two groups, whey and caseins, and they have recently expanded to include a third group, proteins from the MFGM [6,16]. Within these groups, the proportions of whey and casein change throughout lactation, with the ratio of whey to casein being the highest during colostrum and then reaching a steady state in mature milk [9,11,12,16,51], see Table 1 for details. This effect is also attributed to the increase in milk volume throughout lactation [51]. Mucins are found within the MFGM, and therefore are not designated as a percentage of protein, since they are associated within the fat layer and typically make up less than one percent of total protein [6]. Proteins making up the whey fraction of milk predominantly include: α-lactalbumin, lactoferrin, secretory IgA (sIgA), albumin, and lysozyme; and, the casein fraction of milk include: α-S1-, β-, and κ-casein. Unlike bovine milk, human milk does not contain β-lactoglobulin or α-S2-casein, which can trigger immunogenic responses to cow’s milk proteins and then potentially other subsequent food allergies later in life when consumed by susceptible infants [52,53,54,55].

It is often difficult to make comparisons of the protein content in human milk across differing literature sources due to the variability in protein concentrations reported. These variations arise because of the expanse of laboratory methods that are used to determine protein concentration. Examples of differing methods of protein assessment include, in order of low to increasing complexity and accuracy: colorimetric assays, like bicinchoninic acid (BCA); nitrogen determination methods, like Kjeldahl; and, chromatographic methods, like matrix-assisted laser desorption/ionization (MALDI) and nano-spray liquid chromatography tandem mass spectrometry (LC-MS/MS). Mass spectrometry-based methods for the analysis of the human milk proteome are the most ideal, as they allow for both quantitative and qualitative analysis of the entire proteome, proteins with PTMs, like glycosylation, and the endogenous peptidome in one systematic approach.

Clear objectives need to be determined prior to protein analysis to define whether protein assessment is to be done, as true protein measurement vs total nitrogen estimates. This distinction is important, because measurements between the two can differ anywhere from 1–37% [8]. Protein estimated from the measurement of nitrogen often over-estimates the true protein content of human milk because it is assumed that all nitrogen is protein and does not consider the non-protein nitrogen (NPN) compounds [8]. In human milk, protein accounts for approximately 75% of nitrogen-containing compounds [56]. The remaining 25% of the nitrogen in milk is accounted for by NPN compounds, such as urea, nucleotides, endogenous peptides, free amino acids, DNA, and RNA [57,58]. From this, it is estimated that endogenous peptides make up 10–15% of NPN [58]. However, nitrogen studies have not considered glycosylation, and therefore what percentage of nitrogen is coming from this PTM has not be determined. 

## 4. Glycoproteins

Protein glycosylation, the process of adding sugar units to a protein, is one of the most prominent PTMs on human milk proteins. It has been estimated that up to 70% of human milk proteins are glycosylated [59]. Unlike protein sequences, the biosynthesis of glycosylation cannot be directly predicted from the gene. However, the enzymes, including glycosyltransferases, glycosidases, and transporters, which are involved in the process, are directly encoded in the genome [60,61]. A glycoprotein carries one or more glycans covalently attached to a polypeptide backbone, usually via *N*- or *O*-linkages [62,63]. The *N*-glycosylation in human biology is designated as a sugar chain, starting with a *N*-acetylglucosamine (GlcNAc) residue that is linked to an asparagine residue of a polypeptide chain, normally with consensus peptide sequence: Asn-X-Ser/Thr (X is any amino acid except Pro) [64,65]. Additionally, some human milk proteins have *N*-glycans that occur at Asn-X-Cys, e.g., alpha-lactalbumin [66]. Alternatively, *O*-glycosylation is frequently linked glycans via *N*-acetylgalactosamine (GalNAc) to a serine or threonine residue of the polypeptide [67]. The expression level and glycosylation level of glycoproteins in human milk can vary throughout lactation and or biological situations [68]. Protein glycosylation in milk is of special interest, since it is relevant to proteolytic susceptibility, and it functions as competitive inhibitors of pathogen binding and immunomodulators in the gut. As a result, the glycosylated protein in human milk helps to shape the developing gut and immune system of the growing infant [47].

The quantitative variation in expression for some major glycoproteins has been reported, (Table 1). The level of glycosylation, especially the site-specific information throughout lactation, is less detailed in the literature, with the exception of a few proteins e.g., lactoferrin, bile salt activated lipase (BSSL), sIgA, secretory IgM (sIgM), and α-antitrypsin [69]. The less characterized site-specific glycosylation is mainly due to two analytical challenges: (1) the dynamic range of human milk proteins and (2) the complexity of glycosylation. 

The global human milk glycosylation analyses is not often carried out on the intact glycopeptide level with respective site-specific glycosylation information due to these analytical challenges. Instead, analysis has mainly focused on the mapping of *N*-glycosylation sites of enzymatic deglycosylated peptides or on the enzymatically released *N*-glycans, as these analyses circumvent the analytical challenges. Interestingly, the analysis of the released *N*-glycans has revealed that a specific feature of human milk glycosylation is the multiple and abundant *N*-glycan fucosylation, when compared with bovine milk or human serum [70]. In humans, mono-fucosylation tends to be core fucosylation [71], while additional fucosylation is likely terminal fucosylation. Core fucosylation is important, because it can prevent the enzymatic cleavage of proteins from membrane surfaces [72]. The feature of having terminal fucosylation on glycoproteins represents a structural homology to human milk oligosaccharides (HMOs). This is of interest, because, if these two components have similar structures, then perhaps they share similar functionalities, which is not well characterized for terminal fucosylation on glycoproteins. However, it is well described that the fucosylated HMOs are correlated with the increasing diversity of the neonates gut microbiota. Future research should aim to further investigate the glycoproteome and the development of the infant gut. 

## 5. Endogenous Peptides

To date, the literature on the endogenous peptidome in human milk remains limited and there are no studies investigating PTMs of the peptidome, to our knowledge. This is truly astounding when considering that human milk peptides are more diverse than proteins [73]. Additionally, peptides make ideal signaling molecules and they have extreme potential for biomarkers, as they serve as messengers encoding the status of specific regions of the body and potentially the entire organisms as they are less restricted in the movement in relation to proteins. 

Several groups have developed similar MS-based methods to investigate endogenous peptides in human milk [74,75,76,77]. Recently, we established and validated an MS-based method, optimizing peptide extraction, MS fragmentation, and database searching, for a robust and reproducible analysis of endogenous peptides in human milk [78]. Within this method development, we determined that the most optimal workflow for endogenous peptides in human milk fundamentally depended on the depth of analysis desired and the time to be invested. 

An understanding of naturally occurring endogenous human milk peptides verses peptides that are the digestion products of proteins in the infant’s intestinal tract is important. This distinction is required for understanding functionality. Human milk peptidome studies have consistently found peptides abundantly from the casein fraction of milk proteins, where approximately 50% are derived from β-casein [74,75,76,77,79]. In addition to this, other studies have reported no endogenous peptides identified from major whey proteins, such as α-lactalbumin, lactoferrin, or immunoglobulins, [74,75,76,77,79]. Further studies suggested that the proteases in human milk are responsible for the observed proteolytic activity and that this activity is specific and conserved [76]. However, since the whey portion of the proteome is heavily glycosylated, peptides from these proteins are either not degraded by proteases or are not detected due to the binding glycan. Future analysis of endogenous peptides in human milk should aim to address this gap in knowledge. 

Human milk peptides have been shown to have functional properties beyond the sources of amino acids; for instance, they are involved in immunomodulation, opioid-like activity, antioxidant, antimicrobial, and antiviral action, and probiotic action [74,77,80,81,82,83]. The functionality of human milk peptides require that they are hydrolyzed from their parent protein, in which case proteins can be seen as the carriers of functional peptides. Previous studies have suggested that the peptide sequences of human milk are not due to random proteolytic digestion but rater selective proteolysis of the mammary gland [77]. The driving forces for this selectivity may be for protective factors for both the mother and the infant [77]. The benefits of intact peptides for the mother and infant must have served an evolutionary advantage as the energy cost of protein production is high, and therefore it would not be opportune for biological systems to break down these proteins if they did not serve some importance. The proposed health benefits for the mother and infant include the prevention of bacterially induced mastitis and protection against gastrointestinal infections, respectively [77]. Additionally, it has been shown that proteolysis within the mammary gland is increased during times of inflammation or infection [26], leading to altered milk compositions, which can include increased serum proteins and potentially serum derived peptides in milk. Thus, the analysis of the human milk peptidome could provide insights into the maternal-infant health dynamic. 

To date, there is huge interest in determining the functionality of human milk endogenous peptides, which are unique and diverse from that of their parent protein. Particularly, endogenous peptides with a wide range of bioactivities, including antimicrobial, antihypertensive, antithrombotic, and immunomodulatory [84]. While some peptides have established functionalities, other peptides only have proposed functionalities that are based on sequence motif. Additionally, biological insights of endogenous peptides can be achieved by different strategies, such as site visualization, mapping the peptide to the precursor protein back bone; enzymatic mapping, assessing which enzymes released the peptide from the precursor protein; peptide structure predictions, predicting the three-dimensional structure of the peptide from the amino acid sequences and PTMs; and, predicted functionality by homology searches against databases of known functional sequences. Dallas et al. previously reviewed these topics in detail [84]. A major limitation in the analysis of bioactive functionality is that it is reliant on predictive bioinformatics, which is contingent on the quality and completeness of databases currently available. One group has sought to bridge this gap in knowledge by deriving a database specific to milk peptides from various mammalian species with known functionality, Milk Bioactive Peptide Database (MBPDB) [83]. 

## 6. Enzymes

Human milk is comprised of a mixture of proteolytic enzymes, zymogens, protease activators, and protease inhibitors, and therefore the net proteolytic activity is dependent on the quantitative interaction of these components. Additionally, these numerous enzymes all have differing functionalities. The main proteases in human milk include plasmin, trypsin-2, cathepsin-D, neutrophil elastase, thrombin, kallikrein, and several amino- and carboxypeptidases [26]. These proteases are secreted in their inactive form and are then activated by protease activators, such as tissue-type plasminogen activator (t-PA) and urokinase-type activator (u-PA) [26]. The system maintains balance with protease inhibitors, such as α1-antichymotrypsin and α1-antitrypsin [85,86]. The mammary gland is an ideal environment for these proteolytic enzymes, as most of them are capable of functioning at body temperature and they can function at milk’s neutral pH [82]. 

In general, it is well established that human milk contains higher levels of enzymes than bovine milk [87], and further that the enzymes in human milk are inherently different than the same enzyme being expressed in a different body fluid. This difference arises from human milk enzymes having a more highly organized tertiary structure, resulting in greater hydrophobicity, which in turn may account for their resistance to proteolysis and denaturation in the infant’s gastrointestinal system [26,85,86,87,88,89,90]. Studies have shown that the neutral pH of human milk provides a buffering capacity in the infants stomach, which increases the pH and limits proteolysis from pepsin [91]. This increase in the pH of the infant’s stomach is thought to be important for facilitating bacterial colonization of the gut, and that proteolysis is facilitated by milk proteases, which remain active in the infants stomach, such as cathepsin_D and plamin [91]. There are extensive reviews on the enzymes in the mammary gland, human milk, and throughout infant digestion [26,85,86,87,88,89,90,91], for this reason we choose to highlight enzymes that function within the mammary gland to establish the proteome, especially highlighting plasmin, and enzymes that act in secreted milk, which function later on in the infant’s digestive tract. 

### 6.1. Enzymes that Function in the Mammary Gland

Within the mammary gland, enzymatic proteolysis specifically and reproducibly acts as a controlled proteolytic system [88]. Evidence for this can be observed by endogenous peptides, which are characterized by overlapping ladder peptide products that originate from a specific region of the parent protein. These characteristics are due to the nature of the parent protein, PTMs, secondary and tertiary structure, and the abundance in relation to available proteases. It has been identified by Timmer et al. that four key factors determine the specificity of proteolysis: (1) spaciotemporal co-localization, (2) exosite interactions, (3) sub-site specificity, and (4) structural presentation [92]. Guerrero et al. considered these aspects of proteolysis in their analysis of endogenous peptides in human milk and investigated the intrinsic disorder of a protein in relation to the identified peptides [76]. Wherein, the intrinsic disorder is where a part of the protein is “natively unfolded” [93], containing regions that lack stable tertiary structures, which allow for easier proteolytic access and degradation. Overall, in this analysis, it was determined that the observed specificity of endogenous peptides could be explained by the protease activity within the mammary gland. This activity was explained by three factors: (1) the intrinsic disorder of proteins; (2) the advantage of a single cleavage for the generation of peptides at the N and/or C-termini; and, (3) the participation of enzymes with high specificity for peptides being observed at internal regions of proteins [76]. 

It has been previously demonstrated that peptides from major whey proteins, such as lactoferrin, α-lactalbumin, and secretory immunoglobulins, remain intact and they do not contribute to the milk peptidome [74,76,77,84], under healthy conditions. These studies demonstrated that the majority of endogenous peptides were derived from caseins, osteopontin, and polymeric immunoglobulin receptor, and that these peptides represented a minority of the total protein content. Further, these studies hypothesized that this was due to the proteases that were present in human milk, originating in the mammary gland, such as plasmin, cathepsin D, elastase, cytosol aminopeptidase, and carboxypeptidase B2, and that they were active throughout lactation. It is also thought that major whey proteins are not digested in the mammary gland due to a lack of association with micelles and inhibited protease activity, potentially due to PTMs [77]. 

Plasmin is one of the main proteases that hydrolyzes human milk proteins in the mammary gland [77]. It is thought to be an important protease in human milk, as it helps to facilitate the capacity for protein digestion of the infant. Plasmin is a trypsin-like, serine-type protease and it is highly specific for cleavage of peptide bonds of lysine, and to a lesser extent, arginine, at the N-terminal domain. Plasmin activity in milk is controlled by a heterogeneous system of inhibitors and activators. The inactivated zymogen, plasminogen, is converted to active plasmin by two serine proteases, t-PA and u-PA [26]. In addition to these activators are inhibitors of plasmin, including type-1 plasminogen activator inhibitor (PAI-1), inter-α-trypsin inhibitor, and α1-antitrypsin [26]. The inactive form of plasmin, plasminogen is known to associate with the casein micelles in milk [94], as does the activator t-PA. Therefore, the majority of active plasmin in milk is associated with the casein micelle structure. This has been demonstrated in studies that showed β-casein derived peptides that were made up the greatest abundance and relative peptide count, even though it is not the most abundant protein in milk [26,77]. Subsequently, it is also reasonable to expect low amounts of endogenous peptides from whey derived proteins in human milk, due to a lack of association with plasminogen and activator t-PA, and an association with plasmin inhibitors [26,77]. 

### 6.2. Enzymes Present in Milk

The incorporation of proteases in human milk may be an inherent evolutionary design, which is meant to aid in infant digestion. This is important, as infants are born with developmentally naïve digestion systems, starting in the mouth with little salivary proteases and extended to the gastrointestinal system, in which they produce relatively little gastric acid and express low protease activity in comparison to older children and adults. Additionally, the consumption of milk makes the pH of the stomach more neutral than that of an adult. However, even with these known developmental disadvantages, we know that infants are capable of digesting and absorbing milk proteins. It has been hypothesized that this is because proteases are simultaneously delivered via milk to aid in infant digestion [26], which makes sense, taken together with the infant’s low digestive capacity. Evidence for supporting this has been recently reported via peptidomic and bioinformatics analysis that showed milk proteases were actively breaking down protein within the human infant’s gastrointestinal tract [95,96,97]. The concentrations of proteases, their inhibitors, and activators have also been reported to change across lactation. Wherein, the mean protease activity, in milk, is reported to decline throughout lactation, reducing in concentration as the infant becomes more capable of utilizing its own digestive system. It is reported that protease activity increases during involution, and it is thought that this is to aid in returning the mammary gland to pre-gestation status. Dallas et al. recently reviewed these topics extensively [26]. 

The digestibility of the human milk proteome in the infant’s gastrointestinal tract is of importance, because of the consequences of a lack of digestion. For instance, if proteins are not properly digested in the small intestine than this can result in an incomplete amino acid hydrolysis required for growth and would, therefore, increase the infant’s total protein requirements. Further on, if proteins reach the colon, and they are not excreted in feces, they could contribute to over growth of bacteria that are capable of protein-fermentation, such as *Clostridium perfringens* and various *Bacteroides* species [26]. This, in turn, could result in lower populations of beneficial carbohydrate consuming bacteria in the infant gut, such as bifidobacteria [26]. 

## 7. Functionality

The human milk proteome contributes to a wide range of functionalities, including: growth as sources of amino acids; enhanced bioavailability of micronutrients, such as vitamins, minerals, and trace elements; improved cognitive development; immunogenic training as innate and adaptive immunity; and, promoting intestinal growth and maturation via interactions with the microbiome. These functionalities are achieved by varying proteins, endogenous peptides, and their PTMs. Importantly, some proteins are capable of exerting multiple functionalities. One example of this is lactoferrin, which can act as a nutrient transporter, defend against pathogens, stimulator of commensal microbiota, and more [2,3,4,9]. Moreover, the individual components of the human milk proteome are able to synergistically work to drive functionality. Several examples of this exist, including the interaction of lactoferrin and lysozyme on antibacterial functionality and the interplay between sIgA and endogenous peptides from pIgR to sequester bacteria at the mucosal interface in the gut. We break down these functional topics over the next several sections and discuss the major parts of the proteome that contribute to specific functions. 

### 7.1. Growth

Dietary reference intake values for infant protein are based on the protein content of human milk, as this is optimal for meeting the requirements for growth and maintenance. Across the ages of 0 to 4 months, the average protein intake for infants consuming human milk is 8 g/day [98]. The composition of the human milk proteome is one of the major contributing factors to infant growth trajectories. For example, an overall high percentage of whey vs casein proteins corresponds to slowing growth rates in human infants [10]. This is due to an even lower level of casein protein in human milk that already contains one of the lowest concentration of caseins relative to other studied species [51]. Human milk proteins drive infant growth, as they are important sources of essential amino acids after digestion. Moreover, proteins and their derived peptides support infant growth by the enhancement of nutrient absorption and digestion. 

One way digestive enhancement is achieved is by increasing the solubility and bioavailability of nutrients. A well-known example is β-casein, which forms casein phosphopeptides (CPPs) in the mammary gland and during infant digestion and it functions to chelate with minerals and promote the absorption of calcium, zinc, and iron [99,100,101]. Another example is the most prevalent whey protein α-lactalbumin, which has binding sites for calcium and zinc [102,103], which help to increase mineral absorption. The glycoprotein haptocorrin helps to protect the acid-sensitive vitamin B_12_ pass the infants’ stomach, for later absorption in the small intestine [104]. Another mechanism of enhanced nutrient uptake is via protein receptors. For instance, iron that is bound to lactoferrin is not released in the intestine due to the high binding affinity, but for infants, the uptake of iron is increased via the lactoferrin receptor [105]. The lactoferrin receptor is suggested to be the principal iron transport pathway in early life [106], which indicates the nutritional importance of lactoferrin for iron absorption for neonates. Additionally, iron that is provided by lactoferrin is also well utilized in adults [107].

Enzymes in human milk are essential as they contribute to digestion in neonates, as they have rather immature digestive systems and are incapable of producing sufficient quantities of enzymes to fully facilitate digestion [88]. One well-accepted example is bile salt-stimulated lipase (BSSL) which has a wide substrate specificity to hydrolyze mono-, di-, and triglycerides, cholesterol esters, fat-soluble vitamin esters, phospholipids, galactolipids as well as ceramides [108,109], thus aiding in the digestion of milk lipids. [110]. The newborn normally has the ability to digest lactose, the main carbohydrate in human milk, by lactase present in the small intestine [111]. However, the main enzyme for complex carbohydrates, α-amylase, is low at birth [112], but it is supplemented by human milk [113] and it may aid in the digestion of complex carbohydrates when complementary foods are introduced to breastfed infants [114]. 

Cognitive development is another important aspect of growth, however the proteome is often overlooked in this regard and more attention is placed on human milk oligosaccharides and fatty acids. Human milk lactoferrin and proteins of the MFGM have been shown to be important in the cognitive development of the infant [9]. In piglet models, it has been shown that feeding bovine lactoferrin at human milk concentrations was associated with the differential expression of 10 genes in the brain-derived neurotrophin factor (BDNF) signaling pathway and later downstream target proteins of this pathway that are important for neurodevelopment and cognition [115]. Other studies using piglet models have shown that feeding lactoferrin resulted in upregulated intestinal gene expression of BDNF and improved gut maturation, linking the gut-brain-microbe axis in a way not previously reported [116]. While intriguing, these results have not been investigated in the human infant. The only randomized controlled trial (RCT) investigating formula with supplemented MFGM containing 4% protein vs standard formula found that the MFGM supplemented group had significantly higher mean cognitive scores in the Bayley Scales of Infant and Toddler Development when compared to the standard formula group [117]. Moreover, the cognitive scores of the MFGM supplemented group were not different as compared to that of the breast-fed reference group at 12 months of age [117]. 

### 7.2. Immune

The specificity of human milk composition to meet the needs of an individual infant can be exemplified in the immune modulating components, which aim to protect the infant and help to drive the immune development of the infant’s naive immune system. There are a multitude of immune modulating components in human milk, including: antigens, cytokines, immunoglobulins (Ig), polyunsaturated fatty acids, and chemokines [118]; leucocytes, including macrophages, neutrophilic granulocytes, and lymphocytes [119]; immune stimulating proteins and glycoproteins, such as lactoferrin and sIgA [54]; and, additional components, such as hormones, growth factors, and endogenous peptides [120]. The immunogenic activities of the milk proteome are critical for establishing innate immunity and developing adaptive immunity. Additionally, it may play a role to some extent in allergy prevention or development, depending on whether it is acting to build tolerance or potentially cause sensitization. However, many characteristics of this are unknown and current research is aiming to better understand these mechanisms. 

#### 7.2.1. Innate Immunity

The innate immune system is the first line of host defense against pathogens [121] and it is extremely important for infants, since they are lacking mature adaptive immunity. The human milk proteome compensates neonatal innate immunity through many different ways: (1) the inhibition of growth of pathogens; (2) the inhibition of the binding of a pathogen to its receptor; and, (3) regulation of immune response and inflammation. 

One way to inhibit the growth of pathogens is to have competition for the resources required for growth. For example, lactoferrin has a high affinity to bind free ions, which are essential for bacterial growth. By making the free ions unavailable, lactoferrin has a broad bacteriostatic effect. Same holds true for haptocorrin, as the major vitamin B_12_ binding protein, normally unsaturated in human milk, to withhold vitamin B_12_ and inhibit the growth of bacteria [122]. However, haptocorrin does not have a general antibacterial activity, rather its activity was found for a single enteropathogenic *Escherichia coli* (*E. coli*) O127 strain (EPEC) [123] and *Bifidobacterium breve* [124]. 

Another way to inhibit the growth of pathogens is the disruption of membrane structure. Human milk contains a high concentration of lysozyme, which is an enzyme that is capable of hydrolyzing β-1,4 linkages of *N*-acetylmuramic acid (NAM) and *N*-acetylglucosamine (NAG) in bacterial cell walls, leading to the instability of cell wall and bacterial cell death [125]. Lysozyme has been shown to synergistically work with lactoferrin to kill gram-negative bacteria in vitro [126]. With the help of lactoferrin to bind and remove lipopolysaccharide from the outer cell membrane of gram-negative bacteria, the inner proteoglycan matrix is left accessible for lysozyme. Human milk also contains defensins and cathelicidins [127], which have shown to have synergistic antibacterial effects with lysozyme [128]. In addition to the enzymatic activity, human lysozyme is cationic [129] and it can insert into and form pores in negatively charged bacteria membranes [130,131]. 

The membrane attack complex (MAC) as a result of the activation of the complement system is capable of forming pores in the lipid bilayers of bacteria and it leads to cell lysis and death [132]. The complement system is an important part of innate immunity and it also plays a role in adaptive responses [133]. Human milk contains many proteins in the complement system, yet in low levels, although systematic research of the whole system is still lacking. The assessment of individual complement proteins indicate that the complement system in human milk might provide additional immunological and non-immunological protection for infants [134]. 

The membrane structure of pathogens could be damaged and the pathogen growth could be inhibited by a so-called “lactoperoxidase system (LPS)”, which consists of lactoperoxidase and thiocyanate (SCN^−^) which occur naturally in human milk, and H_2_O_2_, which is generated by bacteria [135]. When the H_2_O_2_ is present, lactoperoxidase uses it to oxidase SCN^−^ to hypothiocyanite (OSCN^−^), which has broad antimicrobial activity [136]. Moreover, the overexpressed OSCN^−^ could be eliminated by antioxidants presented in milk to limit the local tissue damage by oxidative stress [5]. Lactoperoxidase is a glycoprotein and it is resistant to proteolysis [137], thus playing a role in infant host defense. 

When pathogens enter the infant’s gastrointestinal tract, the first stage of the infection is colonization by adhesion to host epithelial cells. The intestinal epithelial cells are heavily covered with glycans. Cell surface glycan epitope recognition is the first step of enteric pathogens in their pathogenesis [138]. Milk glycans, in the free form of HMOs or conjugated form in glycoprotein and glycolipid, might have epitopes as part of their structures, and competitively recognize and bind to either the lectin receptors of epithelial cells or pathogen lectin receptors [139]. Thereby, inhibiting the adhesion of a pathogen and infection afterward. Many glycoproteins in human milk have demonstrated the capability to block the interaction of epithelial cells and enteric pathogens, e.g., Tenascin-C for HIV-1 [140]; lactoferrin for *Escherichia coli* O157:H7 and *Salmonella enterica* [141]; BSSL for Norwalk virus [142]; mucins for rotavirus [143], Norwalk virus [142]; and, *Salmonella enterica serovar Typhimurium* SL1344 [144]. One critical aspect in exploring the roles of milk glycoproteins is to characterize the attached glycans and their specificities since the epitope recognition is structure-based. Lactoferrin has a lower ability to interrupt the adhesion of bacteria to epithelial cells when fucose moieties are enzymatically removed [141]. Sialylated glycans of secretory immunoglobulin A (sIgA) were able to bind the S-fimbriated *E. coli* strains and inhibit adhesion [145]. Moreover, mothers have different abilities to produce specific glycan structures [146], as discussed in the glycoproteomics section, as well as attached glycans of glycoproteins [71]. The individual-specific properties add another level of complexity and lead to unique selective inhibition for endemic pathogens.

The immature intestinal mucosa of the neonates is overly sensitive to infection, because of overexpressed inflammatory genes and under-expressed negative feedback regulator genes [147]. Human milk helps to regulate this immunologic balance in neonates, not only by reducing exposure to pathogens and the prevention of infection, but also via modulating the immune response to minimize inflammation pathology for the breastfed infant [148]. Many components in human milk, including proteins/glycoproteins and peptides, have immunomodulatory functions. For example, soluble isoforms of toll-like receptor (TLR)-2 serves as a decoy receptor, suppressing TLR2 activation and decreasing interleukin (IL)-8 and tumor necrosis factor (TNF)-α release, thus reducing inflammation [149]. Glycoproteins, such as CD14 [150], lactoferrin [151], and lactadherin [152], can regulate TLR4 signaling, whereas the endogenous peptide, β-defensin 2, suppresses TLR7 expression [153]. In a recent RCT, bovine osteopontin was added to infant formula and the infants were fed either regular formula or osteopontin supplemented formula [154]. The results from this RCT showed that infants consuming osteopontin supplemented formula had significantly lower serum concentrations of pro-inflammatory cytokine transforming growth factor-α as compared to non-supplemented infants, and moreover that these infants had cytokine profiles more similar to that of breastfed infants [154]. 

#### 7.2.2. Adaptive Immunity

While their own adaptive immune system is maturing, neonates may get protection from the products of the adaptive immune system of the mother, which was first reported in 1892 [155]. One primary protective product is a protein family, called immunoglobulins or antibodies, which come from several sources and they are transported into milk by receptor-mediated processes, reflecting the antigenic stimulation of immunity of the mother [156]. Immunoglobulins include several classes, like IgM, IgA, IgG, IgE, and IgD [157]. In human milk, the most abundant immunoglobulin is sIgA, followed by sIgM and IgG [158]. 

Both sIgA and sIgM are polymeric, typically dimeric IgA and pentameric IgM, which are formed through the covalent interaction with a joining (J) chain and secretory component (SC) from the endoproteolytic cleavage of the polymeric immunoglobulin receptor (pIgR) [159]. sIgA and sIgM are produced in a similar manner and represent the history of the antigenic stimulation of mucosal immunity in the mother [119]. When a pathogen enters the mother’s gut or upper airways, the Peyer’s patch acquires the pathogen and its antigens are presented by M cells to circulating B cells, which migrates to the serosal (basolateral) side of the mammary epithelial cell and it produces IgA and IgM. As the IgA and IgM move from the serosal to the luminal (apical) side of the mammary epithelial cell, they are glycosylated and complexed to form sIgA and sIgM, which are secreted into milk [160]. When the infant consumes milk, the sIgA and sIgM associate with free SC and they are resistant to digestion [161]. Additionally, free SC is responsible for intracellular neutralization of some viruses and it is capable of binding bacterial components, like *Streptococcus pneumonia*-derived SpsA protein and prevent the invasion of epithelial cells [161]. sIgA and sIgM both carry the memory of pathogens faced by the mother, which allows for them to provide the same protection to the infant by binding to recognized pathogens and inhibiting their ability to infect the infant [162]. This mechanism is called the enteromammary link, where maternal immunity is transferred to the breastfed infant [163]. Milk sIgA is considered to be the dominant immunoglobulin to protect mucosal surfaces of infants from pathogens and enteric toxins, through intracellular neutralization, virus excretion, and immune exclusion [158,164]. Additionally, sIgM has been demonstrated to activate the complement cascade in vitro [165].

IgG is required to provide systemic immunity and it is transferred before birth. During pregnancy, IgG is transferred from mother to the fetus via transplacental passage and it provides crucial protection to the neonate in the first weeks of life after birth [166]. The main portion of human milk IgG is transported from serum through neonatal Fc receptor (FcRn) [167], and its four subclasses are ordered in cord serum level as IgG1, IgG2, IgG4, and IgG3 [168]. The concentration of IgG in human milk is much less than sIgA since the human mammary gland has considerably less FcRn than pIgR. Additionally, IgG is less resistant to digestion than sIgA. The remaining intact IgG in the intestinal lumen might be involved in immune surveillance by binding antigens and enhancing the local mucosal immune response by transporting the IgG-antigen complexes into *lamina propria* for the subsequent induction of immune activation or tolerance [169]. On the other hand, maternal antibodies may inhibit infant vaccine response depending on the ratio of maternal antibodies to the vaccine antigen [170], although maternal vaccination, such as pertussis and influenza, showed protective effects for infants [171,172]. 

#### 7.2.3. Potential Allergens in Human Milk and Immunity

The presence of non-human proteins in human milk was reported many decades ago [173], especially some potential food allergens, including proteins from cow’s milk, eggs, peanuts, and wheat, in the form of degraded peptides, intact proteins, and/or an immune complex with antibodies [14,174,175,176,177,178,179]. Besides food proteins, other sources are also reported to be present in human milk, e.g., house dust mite [180]. It is still not quite clear how foreign proteins enter human milk and the consequences for infant health. It is difficult to elucidate the exact roles of foreign proteins in a complicated mixture as human milk, where pro-inflammatory and anti-inflammatory factors are both present and change constantly [53]. However, some critical reviews have revealed that breastfeeding positively promotes the development of tolerance in the infant and protects against allergic diseases, asthma, and atopic dermatitis [53,181,182]. Moreover, studies have shown that avoidance of food allergens in maternal diet during lactation, or postponed introduction of risky foods in children have not shown a clear benefit [183]. Instead, introducing a food antigen in early life with low levels of continuous exposure might reduce the risk of developing related allergies [184]. The potential allergens introduced in human milk by months of breastfeeding might be the ideal way to promote tolerance [185]. Tolerance to harmless antigens plays a crucial role in immune homeostasis by preventing potentially dangerous hypersensitivity reactions [186]. 

### 7.3. Gut Development

The infant’s immune system and gut develop concurrently. As the infant’s immune system is immature at birth, so is the mucosal lining of the gastrointestinal tract and the gut with limited microbiota. The maturation of the gut during infancy relies, extensively, on adequate nutritional support, and it is shaped by the delivery of human milk bioactive constituents. The human milk proteome plays a large role in this maturation by providing support to intestinal epithelia, the mucosal lining and gut microbiota for developmental programming [187]. Moreover, intestinal tropism is primarily derived from human milk protein, rather than by lipid or carbohydrate [187]. The bacteriostatic properties of the milk proteome help to create an environment within the infants’ gastrointestinal tract, where unrestrained bacterial growth is prohibited and the microorganisms are removed from the small intestine without causing inflammation [3]. In turn, this contributes to the development of a healthy microbiome for the infant [3]. All together, the milk proteome results in enhanced responsiveness of the intestine to dietary, physiological, and pathological challenges [187].

The infants gut physiology consists of intestinal epithelial cells (IECs) covered by a mucus layer, and it is at this interface that innate and adaptive immunities must cooperatively function to protect the infant from the vast assault of stimuli [161]. Intestinal homeostasis is maintained by IECs, which act as a physical barrier driving synergistic immunity against invading pathogens. The main player of the IECs is the polymeric immunoglobulins (pIgs). As described in the adaptive immunity section, pIgR is complexed to dimeric IgA (dIgA) by the J-chain; this is a critical step in the translocation of IgA across the IECs and secreted into the lumen, where sIgA can exert its protective functions. This complex formation and translocation occurs both in mammary epithelial cells and in IECs. Once this dIgA-pIgR complex is expressed on the apical surface of IECs, it is released by proteolytic cleavage. Research studies have shown that pIgR cleavage occurs on the cell surface and not inside the cells, which indicates that this is a highly localized process [161]. However, the enzyme responsible for the cleavage of pIgR has not been identified, although some research suggests that the enzyme is a serine proteinase [161]. Maternal sIgA in breast milk is supplied to the infant in sufficient quantities to promote gut health and maturation, as the infant is incapable of producing sufficient quantities of sIgA until approximately two years of age. After sIgA is translocated from the IECs it enters the intestinal mucus layer where it helps to maintain spatial segregation between the microbiota and epithelial surface [188]. It has been postulated that the production of sIgA and pIgR is regulated by the gut microbiota by products that are shed from the microbial community [188]. This, in turn, helps to stimulate a commensal relationship between the hosts gut and microbiome, in which the human milk proteome is a key driver for the developing infant. 

Mucosal development is a major determinant of infant’s gut health and homeostasis. Mucus producing goblet cells, Paneth and M cells in the intestine, function beyond nutrient absorption and have a major role in bacterial mucosal cross talk and innate barrier function [187]. These cells are responsible for the secretion of mucus and mucin proteins, which are important in the nonspecific protection of the gut lining [187]. Mucus in the gut lining is a complex gel that is primarily composed of water and electrolytes, but it also contains many proteome components, such as mucins, glycoproteins, immunoglobulins, albumin [187], and bioactive peptides. It is at the mucus interface that a commensal microbiome can be established, while pathogenic bacteria can become trapped and acted upon by the functional proteome. 

Many human milk proteins play a part in this, for instance, the sIgA-pIgR complex, as just described. Another protein, κ-casein, which is heavily glycosylated with glycan structures that are similar to exposed glycans on mucosal surfaces, helps to prevent pathogen binding. This protective functionality is most likely occurring from the proteolytic product of κ-casein, glycomacropeptide (GMP), which binds to mucosal surfaces, thereby preventing the attachment and infection of pathogens [4]. Similarly, haptocorrin may act in the fashion as κ-casein in inhibiting pathogen interaction at the mucosal surface.

Proteins that are associated with the MFGM are important for gut development, such as mucin-1, xanthine oxidoreductase, butyrophilin, CD36, adipophilin, lactadherin, and fatty acid-binding protein [3,4,9]. These proteins help to stimulate intestinal epithelial health, promote gastric stability, and contribute to the antiviral and antibacterial activities in the infant gastrointestinal tract [3,4,9]. Specifically, mucin-1 has been shown to inhibit the invasion of *Salmonella typhimurium,* in a model of fetal intestinal cells, at concentrations that are similar to that of human milk [144]. 

Recent research has investigated the proteome as a cipher for functionally encrypted peptides that are released upon infant digestion. As this review aimed to assess the human milk proteome and endogenous peptides and not the products of digestion, we did not address these current topics. However, we wanted to point readers in the right direction, as these functional peptides are described to exert their bioactivities directly in the gastrointestinal lumen and at peripheral organs after being absorbed at the intestinal mucosa. Please see a recent review by Y. Wada and B Lonnerdal [80], and a few experimental studies [95,96,97]. 

Some proteins have been hypothesized to be more beneficial for the infants gut as intact, such as lactoferrin, lysozyme, α-lactalbumin, and immunoglobulins due to their important functions as antimicrobial and immunologic, and for the presence of the glycan groups. An overall lack of digestion of these proteins may then be functionally beneficial for helping to drive the establishment of commensal gut microbiome, as it is commonly observed that these proteins can be found back intact in infant feces [2,4]. The glycans that are released from these undigested glycoproteins by microbial glycosidases are bifidogenic [189]. Further, it has been suggested that the degree of glycan diversity and rate of glycan site occupancy of these glycoproteins changes to meet the developing gut microbial diversity [141].

## 8. Conclusions

Human milk has historically been known to prevent high morbidity and mortality rates of infants in the first months of life. This is, in large part, due to the human milk proteome, which helps the immature infant fight against infectious diseases, such as otitis media, respiratory tract infections, gastrointestinal infections, and more. Additionally, breastfeeding, in general, has been shown to reduce the risk of development of maternal cancer and the proteome in particular has been shown to reduce the risk of mastitis. The power of the human milk proteome is derived from the enteromammary transfer of maternal immunity and its broad and synergistic functionality, Figure 2. Moreover, the protective and developmental proteome is unique to each individual mother-infant pair. 

There is an overall need for a more comprehensive characterization and quantification of the human milk proteome throughout lactation. Preferred methodologies, such as modern MS, should be considered, especially for human milk, where the complex matrix can cause unwanted complications in other types of analysis, to fully characterize the proteome and derive deeper understandings of its functionality. These current gaps in knowledge could be overcome by some of the topics covered throughout this review, such as quantitative MS to derive concentrations of individual proteins and endogenous peptides, and advanced MS methodologies to better characterize PTMs, like glycans. The complexity of the human milk proteome and its corresponding functions is a challenging but demanding topic for personalized nutrition, as it is responsible for driving infant growth and development, and it even has the potential for the discovery of novel biomarkers and therapeutics. 

## Figures and Tables

**Figure 1 nutrients-11-01834-f001:**
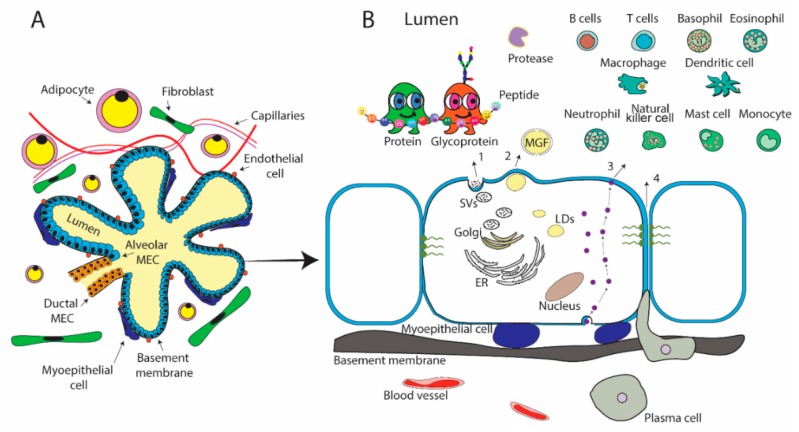
Physiology and transport of the proteome in a lactating mammary gland. (**A**) One lobulo-alveolar cluster connected to a lactiferous duct is depicted. The alveolus cluster, made up of a monolayer of polarized alveolar mammary epithelial cells (MEC) surrounding the lumen, and is connected to the lactiferous duct that is surrounded by a bilayer of ductal MEC. The alveolar MEC are surrounded by basement membrane and a single layer of polarized myoepithelial cells that contract to stimulate milk ejection from the lumen. The alveoli are embedded in a stoma of vascularized connective-tissue that contains lipid-depleted adipocytes, fibroblasts, endothelial cells and capillaries. (**B**) A zoomed in representation of the alveolar MEC is depicted to show the four key transport pathways of the milk proteome. Pathway (1) The exocytotic pathway is the dominant way for the secretion of endogenously generated proteins. These proteins and other aqueous components of milk are transported in secretory vesicles (SVs). Pathway (2) Secretion of lipid-associated proteins by the formation of lipid droplets (LDs), formed in the endoplasmic reticulum (ER), that move to the apical membrane to be secreted as milk fat globule (MFG). MFG are excreted by budding and are enwrapped by the apical plasma membrane of the MEC and become MFGM. Pathway (3) The vesicular transcytosis of proteins from serum or stromal cells. Pathway (4) Direct transport via the paracellular pathway for serum substances and cells, such as immune cells and stem cells. This route of transport is only available during pregnancy, early lactation prior to tight junction closure of MEC, involution, during times of inflammation or preterm birth.

**Figure 2 nutrients-11-01834-f002:**
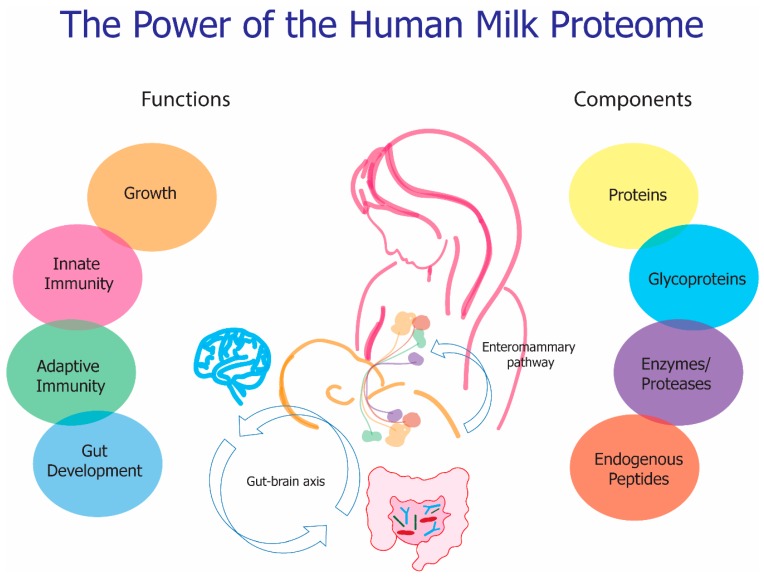
The power of the human milk proteome stems from an evolutionary advantage of cross communication between the mother-infant dyad, via the enteromammary pathway, which in turn stimulated the rise of the uniquely functioning proteome and all of its components. These components work both individually and synergistically to provided protective immunity and overall development to the infant while maintaining health benefits for the mother as well.

**Table 1 nutrients-11-01834-t001:** Overview of reported human milk protein concentrations over lactation and functionality.

**Proteins**	**Protein Name**	**Total**	**Colostrum**	**Early**	**Transitional**	**Mature**	**References**	**Function**
	Total protein	203–1752	360–1690	606–1675	203–1752	362–1632	[16]	
	Total caseins	19–591	42–507	103–355	87–591	19–743	[16]	
	Ratio whey/casein		90:10	78:22	72:28	60:40	[9,16]	
	**Whey Proteins**							
	α-Lactalbumin	275–372	300–560	NA	420	275–372	[17,18]	Lactose synthesis
	Lactoferrin	97–291	291	NA	180	97	[17,19]	Antimicrobial; Gut development
	Osteopontin	6–149	149	NA	NA	6–22	[3,20]	Cell adhesion
	sIgA	22–545	545	NA	150	22–130	[17,18]	Adaptive immunity
	IgG	2–7	NA	NA	5	2–7	[18,21]	Adaptive immunity
	sIgM	1–3	NA	NA	12	1–3	[18,21]	Adaptive immunity
	Lysozyme	3–110	32	NA	30	3–110	[17,18]	Antimicrobial
	α1-Antitrypsin	2–5	NA	NA	NA	2–5	[21]	Protease inhibitor
	Serum albumin	35–69	35	NA	62	37–69	[18]	Transport
	Lactoperoxidase	70 *	NA	NA	NA	70 *^,#^	[22]	Antimicrobial
	Haptocorrin	70–700 *	NA	NA	NA	70–700 *	[3]	Vitamin B12 transport
	Complement C3	11–12	NA	11	NA	12	[23,24]	Innate immunity
	Complement C4	5	NA	5	NA	5	[23,24]	Innate immunity
	Complement factor B	2	NA	2	NA	NA	[23]	Innate immunity
	**Casein Proteins**							
	β-casein	4–442	4–364	18–204	6–414	5–442	[17]	Calcium transport
	α-S1-casein	4–168	12–58	15–85	9–110	4–168	[16]	Calcium transport
	κ-casein	10–172	25–150	47–134	10–172	10–134	[16]	Calcium transport
	**MFGM Proteins**							
	Mucin 1	13–294 ^§^	NA	NA	13–250 ^§^	35–294 ^§^	[25]	Growth promoter
	Lactadherin	3–33 ^§^	NA	NA	4–33 ^§^	3–13 ^§^	[25]	Cell adhesion
	Butyrophilin subfamily 1	500–10,000 *^,^^§^	NA	NA	800–8200 *^,^^§^	500–10,000 *^,^^§^	[25]	Regulation of immune response
	Bile salt-activated lipase	10–20	NA	NA	NA	NA	[3]	Lipid digestion
**Enzymes**	Total protease activity	0.76–1.38 ^†^	1.38 ^†^	NA	NA	0.76 ^†^	[26]	
	Thrombin	7100 **^,^^§^	NA	NA	NA	7100 **^,^^§,^^#^	[27]	Coagulation
	Plasmin	14600 **^,^^§^	NA	NA	NA	14,600 **^,^^§,^^#^	[27]	Proteolysis
	Elastase	200 **^,^^§^	NA	NA	NA	200 **^,^^§,^^#^	[27]	Proteolysis
**Hormone peptides**	Total endogenous peptides	1–2	NA	NA	NA	NA	[3]	
	Ghrelin	7–16 **	6–9 **	NA	7–10 **	13–16 **	[28,29]	Appetite stimulator
	Leptin	16–194 **	16–700 **	NA	20–84 **	165–194 **	[28,30,31]	Energy regulator
	Epidermal growth factor	4–5 **	NA	NA	NA	4–5 **	[28]	Stimulates magnesium reabsorption
	Insulin-like growth factor-1	6–12 *	NA	NA	NA	6–12 *	[28]	Insulin regulator and growth-promoting
	Adiponectin	420–8790 **	NA	NA	661–2156 **	420–8790 **	[31,32]	Glucose and fat regulator
	Parathyroid	1029–5840 ^‡^	1029 ^‡^	4584 ^‡^	5840 ^‡^	NA	[33]	Epidermis development

Notes: Values were obtained from studies analyzing term milk samples unless otherwise noted. MS data from literature was considered first and values are indicated in blue, if no MS data was available, immune assays were used and are indicated in black. Total values were derived from the ranges observed across lactation from all referenced literature. Lactational stages designated as, colostrum: ≤2 days, early: 3–5 days, transitional: 6–15 days, mature: ≥16 days. ^#^ wide range of lactational days reported and values were designated as mature. Values were reported as ranges unless only a single value was designated. All values were rounded to the nearest whole number and are in mg/100 mL, unless designated as * ug/100 mL ** ng/100mL. ^†^ Units for total protease activity are µmol tyrosine/1000 mL/min. ^‡^ Units for parathyroid hormone related protein were in units of pmol/L, only mean values from reference were reported. Most articles do not distinguish between sIgA and IgA or IgA1 vs. IgA2, we therefore choose to represent all values as sIgA. ^§^ Enzymes and MFGM proteins were from preterm human milk samples, MFGM values from paper designated as <15 days were considered transitional. Values from hormone peptides were reported only from normal weight infants. Functions were determined by UniProt assignments based on Protein ID.

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
