# Peer review of "The Functional Power of the Human Milk Proteome"

_nutrients, 2019, doi:10.3390/nu11081834_

Round 1

Reviewer 1 Report

The paper under evaluation focus on the human milk proteome and aimed to present this aspect in a comprehensive way. It's very current and needed topic to cover. 

However it's rather narrative review not systematic review which is methodology weakness. 

The paper lacking of methodology section so we really don't know how the searching to cover the topic was done. In spite of it is not obligatory,  I recommend to add the section Methods with clear description of searching strategy ( keywords, terms of published publication included into review). Indication that this is a narrative review should be included in this section. 

Moreover , It’s better to use simple , scientific langue  instead of  colloquial langue.
 For example  at Row 493 instead of “ competitively plunder” use verb competition , Row 617  “hand to hand”  , use in parallel . Please revise your paper in this context with a  taking attention on subheading. 

Below are more points to consider: 

Row 68 indicate reference 

Row 96 indicate reference and describe mechanism more precisely 

Additionally, the overall proteome can be impacted by inhibitors and activators, present within the mammary gland, and throughout the body allowing for constituents of the proteome to be translocated to the mammary gland via circulation. 

Figure 1 

Unify the numeric name of pathway of milk protein synthesis ( 1-4 or I – IV) 

Row 308 eliminate “-“ 

Row 321 eliminate one dot “ . ” 

From 335 - 444 unify the font size 

Row 468  As I know it is the only one RTC with MFGM supplementation of formula so it’s better to state the only rather that the recent especially that this research was done five years ago. Please verify. 

Row 500 Unify the font size 

Row 477 -482 It’s extremely long sentence, I recommended divide it to separate two sentence. 

Row 487, Row 557 , Row 598 – Following the  Instructors for Author’s with numbering the paragraph – it’s better to use 8.2.1 that 8.2a and 8.2.b ect. 

Row 515  Try to avoid such many repetition in one sentence – I suggest to use phrase:  additional/ further  immunological and non-immunological protection for infants instead of complementary immunological and non-immunological protection. 

Section 8.2 c Please unify the font size - it seems to be a bigger that Section before ( 8.2b  and after  8.3. Follow the Instruction for Authors about numbering.

Row 624 -627 The sentences is too long , hard to understand , made it simplest by divided
Row 630 Please remove double that in this sentence 
Row 631. What does mean “ vast barrage of stimuli” is this contents , are you mean  multiplicity of stimuli ? 
Row 649 What does mean “ major part” , I suggest use noun "determinant"  ? 

Author Response

Thank you for your kind comments and your thoughtful suggestion. Please see our full reply in the attachment. 

Reviewer 2 Report

I would like to thank the authors for their efforts in this review and kindly suggest the following attach comments.

Author Response

Thank you for your kind comments and your thoughtful suggestion and points of discussion. Please see our full reply in the attachment. 
